# Sensitivity Enhancement of Thermometry in Tb^3+^-Doped KY(CO_3_)_2_:Sm^3+^ by Energy Transfer

**DOI:** 10.3390/molecules30040767

**Published:** 2025-02-07

**Authors:** Shijian Sun, Jian Qian, Zheng Li, Lei Huang, Dechuan Li

**Affiliations:** 1School of Physics and Electronic Information, Huaibei Normal University, Huaibei 235000, China; sunsj_0105@163.com (S.S.); qianjianwyyx@163.com (J.Q.); hbnu991210@163.com (L.H.); 2Anhui Province Key Laboratory of Intelligent Computing and Applications, Huaibei Normal University, Huaibei 235000, China; 3Anhui Province Key Laboratory of Pollutant Sensitive Materials and Environmental Remediation, Huaibei 235000, China

**Keywords:** thermometry, Sm^3+^, Tb^3+^, phosphor

## Abstract

Sm^3+^ and Tb^3+^ co-doped KY(CO_3_)_2_ temperature sensing materials were prepared via the hydrothermal method. X-ray diffraction results confirmed the monoclinic phase in KY(CO_3_)_2_:Sm^3+^,Tb^3+^ samples. In this KY(CO_3_)_2_ host, Tb^3+^ transfers energy to Sm^3+^ through cross-relaxation. Notably, a 20 mol% concentration of Tb^3+^ increases the emission intensity of Sm^3+^ by 7.1 times. The fluorescence emission intensities of ^5^D_4_ (Tb^3+^) and ^4^G_5/2_ (Sm^3+^) vary significantly with temperature. Both Sm^3+^-Sm^3+^ and Tb^3+^-Sm^3+^ pairs act as effective emission centers in KY(CO_3_)_2_:Sm^3+^,Tb^3+^ for optical temperature measurement. The relationship between fluorescence intensity ratio (I_542_/I_567_) and temperature reveals that the maximum absolute sensitivity and relative sensitivity of KY(CO_3_)_2_:Sm^3+^,Tb^3+^ are 0.031 K^−1^ and 0.46%K^−1^ at room temperature of 298 K, respectively. In contrast, KY(CO_3_)_2_:Sm^3+^ has a maximum absolute sensitivity of only 0.00051 K^−1^ and a relative sensitivity of 0.11%K^−1^ at 498 K. These results highlight the significant role of Tb^3+^ in enhancing Sm^3+^ emission intensities, making Tb^3+^ doped KY(CO_3_)_2_:Sm^3+^ a promising candidate for thermometry.

## 1. Introduction

Sm^3+^ is an important rare earth luminescent ion that can emit orange light and red light, corresponding to its excited state energy level of ^4^G_5/2_. It is commonly utilized in the synthesis of high color-rendering index white light [1,2] and temperature sensing [3,4]. The effective excited states of Sm^3+^, ^4^G_5/2_ [5], ^4^F_3/2_ [6], and ^4^G_7/2_ [7] allow excited state electrons to transition to the ground state for photon emission. However, the efficiency of electron conversion varies among these states [8], leading to the loss of excited state electrons and, consequently, low fluorescence quantum efficiency. The orange-red light from Sm^3+^ is typically used to enhance the red component in white LED lamps for lighting [9]. However, as the temperature increases, the emission intensity of Sm^3+^ will weaken. The primary reason is that the energy difference between the ^4^G_5/2_ and ^4^F_3/2_ states is approximately 908 cm^−1^ [10]. This allows for thermal excitation to redistribute the other excited state electrons. The redistribution of excited state electrons between thermal coupling energy levels follows the Boltzmann distribution [11]. The emission intensity of Sm^3+^ relative to temperature can be used to determine the emission temperature of luminescent materials. Consequently, fluorescent powders containing Sm^3+^ are also employed as temperature-sensing materials.

In thermometry, Sm^3+^ ions are frequently employed for high-temperature environments. Temperature measurement can be achieved by exploiting thermal coupling between two high-energy states and one low-energy state. In YNbO_4_:Sm^3+^ fluorescent powder, the emission ratio of the high-energy ^4^F_3/2_ and ^4^G_5/2_ states to the low-energy ^6^H_5/2_ state yielded a maximum absolute sensitivity of 0.0007 K^−1^ at 700 K [12]. In GdVO_4_:Sm^3+^, Nikolic et al. reported a maximum absolute sensitivity of 0.00045 K^−1^ at 750 K using the same energy levels [13]. An alternative method involves measuring temperature through emission transitions from a high excited state to two lower excited states. Lojpur et al. achieved a maximum absolute sensitivity of 0.0036 K^−1^ at 580 K in Lu_2_O_3_:Sm^3+^ by analyzing transitions from the excited state ^4^G_5/2_ to the low-energy states ^6^H_5/2_ and ^6^H_9/2_ [14]. Klimesz et al. used the transition from the high-energy state ^4^G_5/2_ to the low-energy states ^6^H_5/2_ and ^6^H_7/2_ in oxyfluorotellurite glasses, achieving a maximum sensitivity of 0.0031 K^−1^ at 700 K [15]. Therefore, those studies demonstrate that both upper and lower energy levels of Sm^3+^ can effectively be used for temperature measurement.

The fluorescence emission ratio of dual emission centers was proposed to enhance the sensitivity of Sm^3+^ ion temperature sensing. In the Ba_3_(VO_4_)_2_:Sm^3+^ system, the maximum absolute temperature sensitivity of 0.039 K^−1^ at 463 K can be achieved using the fluorescence intensity ratio of VO_4_^3−^ and Sm^3+^ [16]. In the Tb^3+^-Sm^3+^ co-doped NaLa(MoO_4_)_2_ system, a maximum absolute sensitivity of 0.02 K^−1^ at 443 K was determined using the thermal coupling energy levels between Sm^3+^ and Tb^3+^ [17]. Similar methods have been reported in Eu^3+^/Sm^3+^ [18], Mn^4+^/Sm^3+^ [19], and Bi^3+^/Sm^3+^ [20] systems. Thus, the fluorescence intensity ratio of Sm^3+^ in dual ion emission systems can enhance the development of high-sensitivity thermal sensing phosphors for low-temperature measurements.

KY(CO_3_)_2_ (labeled as KYC) crystals are excellent optical materials [21]. Y^3+^ ions in the KYC lattice can be substituted by other rare earth ions with similar ionic radii to create multiple luminescent centers. Tb^3+^ doped KYC has demonstrated a fluorescence quantum efficiency of up to 177% [22]. The efficient luminescence of Tb^3+^ ions is expected to enhance the fluorescence intensity ratio of dual ion emission to obtain larger temperature sensitivity values. Additionally, Tb^3+^ and Sm^3+^ ions can facilitate effective energy transfer due to their similar energy levers [23,24]. A smaller difference in energy levels results in a greater influence of phonons on energy transfer at the same temperature. The interaction between Sm^3+^ and Tb^3+^ in KYC could significantly improve the temperature sensitivity of sm^3+^, especially at room temperature. Therefore, this work incorporates Tb^3+^ and Sm^3+^ ions into the KYC system to explore the spectral characteristics, energy transfer, thermal quenching mechanisms, and temperature sensing properties of KYC:Sm^3+^,Tb^3+^.

## 2. Results and Discussion

### 2.1. Crystal Structures

The X-ray diffraction (XRD) patterns of KY_1−*x*_(CO_3_)_2_:*x*Sm^3+^ (*x* = 0, 0.1, 0.2) and KY_0.9−*x*_(CO_3_)_2_:0.10Sm^3+^,*x*Tb^3+^ (*x* = 0, 0.02, 0.10, 0.20, 0.30, 0.40, 0.50) are illustrated in Figure 1. Figure 1a,b shows that the main diffraction peaks for KYC:Sm^3+^ and KYC:Sm^3+^,Tb^3+^ are located at similar positions, with only minor differences in peak intensity. The positions of diffraction peaks in the fluorescent powder align with the standard card (JCPDS: 01-088-1423), and no additional diffraction peaks were detected. The reason is that Tb^3+^ (1.040 Å), Sm^3+^ (1.079 Å), and Y^3+^ (1.019 Å) are all trivalent ions with comparable ionic radii [25]. Tb^3+^ and Sm^3+^ ions can easily substitute for Y^3+^ ions in KYC to form the monoclinic phase KYC:Sm^3+^ and KYC:Sm^3+^,Tb^3+^. When Sm^3+^ is used alone as a doping ion, its concentration in KYC:0.20Sm^3+^ can reach 20 mol% (Figure 1a). When Tb^3+^ ions are co-doped in KYC:0.10Sm^3+^, the concentration of Tb^3+^ can reach 50 mol%. KYC:0.10Sm^3+^,0.50Tb^3+^ can still maintain its original monoclinic structure (Figure 1b).

Figure 1c,d illustrates the Rietveld refinement results for KYC:0.1Sm^3+^ and KYC:0.10Sm^3+^,0.20Tb^3+^ analyzed using FullProf software. The figures show that most diffraction peaks are well-fitted, although there are a few exceptions. In Figure 1c, the (002) and (−221) crystal planes show discrepancies, while in Figure 1d, the (−111) and (−352) crystal planes are less accurately fitted. Furthermore, the fitting quality for the diffraction peaks in Figure 1c is superior to that in Figure 1d. The corresponding fitting parameters are summarized in Table 1. The fitted cell volume reveals that KYC:0.1Sm^3+^ has a greater value than KYC:0.10Sm^3+^,0.20Tb^3+^. This difference is primarily due to the fact that the radius of Sm^3+^ is larger than that of Tb^3+^, which is, in turn, larger than that of Y^3+^.

### 2.2. Microstructure and Elements Distribution

Figure 2 shows the microstructure and energy spectrum of KYC:0.10Sm^3+^,0.20Tb^3+^ samples. As shown in Figure 2a, the KYC doped with Sm^3+^ and Tb^3+^ displays distinct, regular shapes and a monoclinic appearance, consistent with the results obtained from X-ray diffraction analysis. The grain size is different, with a maximum size of about 100 μm. Larger grains may fracture into smaller pieces due to stress during the growth process. The elemental surface distribution of block-shaped grains is illustrated in Figure 2b–e. The graph shows that potassium (K), yttrium (Y), terbium (Tb), and samarium (Sm) are evenly distributed across the grains. Notably, the distribution density of K and Y is particularly high, followed by Tb, while Sm has the lowest density. According to the elemental content in Figure 2f, the atomic percentages of Tb and Sm are 0.7% and 0.3%, respectively, which is close to a 2:1 ratio. Therefore, the elemental content in the KYC:0.10Sm^3+^,0.20Tb^3+^ samples prepared via the hydrothermal method is nearly consistent with the original stoichiometric ratio.

### 2.3. Luminescence Spectra

KYC:0.10Sm^3+^, KYC:0.20Tb^3+^, and KYC:0.10Sm^3+^,0.20Tb^3+^ were prepared to investigate the luminescence characteristics and the energy transfer between Tb^3+^ and Sm^3+^. As shown in Figure 3, when the monitoring wavelength is set to 596 nm, the excitation spectrum of KYC:0.10Sm^3+^ displays a series of narrowband excitation peaks ranging from 300 to 500 nm. Those peaks originate from the electronic transitions of Sm^3+^ ions between the 4f–4f energy levels. The intense excitation peaks occur at 374 nm (^6^H_5/2_ → ^6^P_7/2_) and 401 nm (^6^H_5/2_ → ^4^F_7/2_) [26]. When the monitoring wavelength is adjusted to 542 nm, the excitation spectrum of KYC:0.20Tb^3+^ consists of two parts: the strong absorption bands located in the range of 210–300 nm and a series of small absorption peaks at 310–390 nm. The two strong absorption peaks in broadband are located at 247 nm and 279 nm, respectively, which originate from the energy level transition of Tb^3+^ ions from 4f^8^ to 4f^7^5d_1_ [27]. The linear absorption peaks in the 310–390 nm range correspond to the 4f^8^ → 4f^8^ forbidden transition of Tb^3+^ ions, with sharp excitation peaks at 318 nm (^7^F_6_ → ^5^H_7_), 351 nm (^7^F_6_ → ^5^L_9_), 368 nm (^7^F_6_ → ^5^L_10_), and 376 nm (^7^F_6_ → ^5^G_6_) [28,29]. In the overall excitation spectrum, there is an overlap in the 340–385 nm range between the excitation spectra of KYC:0.10Sm^3+^ and KYC:0.20Tb^3+^, indicating a potential energy transfer pathway between Sm^3+^ and Tb^3+^ ions. In the excitation spectrum of KYC:0.10Sm^3+^,0.20Tb^3+^, the excitation intensities at 401 nm and 374 nm are notably high.

Figure 4 illustrates the luminescent properties of Tb^3+^ or Sm^3+^ doped KYC luminescent materials. Figure 4a displays the emission spectra of KYC:*x*Sm^3+^ (*x* = 0.05, 0.07, 0.10, 0.13, 0.15, and 0.20) at various concentrations of Sm^3+^ doping. The graph illustrates that the emission intensity of Sm^3+^ initially increases and then decreases as the doping concentration changes. At the strongest emission peak of 596 nm, it is evident that Sm^3+^ exhibits the highest emission intensity at a doping concentration of 0.10. The quenching concentration of Sm^3+^ in KYC:*x*Sm^3+^ is greater than 5% in LiSrYW_3_O_12_:Sm^3+^ [11], 1.5% in LiSr_(1−3/2*x*)_VO_4_:*x*Sm^3+^ [30], and 2.5% in LiBaPO_4_:Sm^3+^ [31]. A higher concentration of Sm^3+^ can lead to better emission intensity.

Figure 4b displays the emission intensity of Sm^3+^ ions at 596 nm with varying concentrations of Tb^3+^ in the KYC:0.10Sm^3+^,*x*Tb^3+^ (*x* = 0, 0.02, 0.10, 0.20, 0.30, 0.40, and 0.5) samples. Under excitation at 374 nm, KYC:0.1Sm^3+^,*x*Tb^3+^ exhibits two emission characteristic peaks corresponding to the Tb^3+^ and Sm^3+^ ions. When the Tb^3+^ ion doping concentration is low, the emission intensity of Sm^3+^ gradually increases with the increase of Tb^3+^ ion concentration. The peak intensity of Sm^3+^ at 596 nm reaches its maximum when the Tb^3+^ doping concentration is 0.20. This enhancement in emission is attributed to energy transfer between Tb^3+^ and Sm^3+^. Excited state electrons can be transferred from Tb^3+^ to Sm^3+^ to enhance the emission of Sm^3+^. The inserted graph shows that Sm^3+^ achieves its highest emission intensity, 7.1 times greater than that at a Tb^3+^ ion concentration of 0, at a Tb^3+^ concentration of 20%. However, as the doping concentration of Tb^3+^ ions continues to rise, the emission intensity of KYC:0.1Sm^3+^,*x*Tb^3+^ gradually decreases due to Sm^3+^ concentration quenching.

The concentration quenching of KYC:Sm^3+^,Tb^3+^ can be analyzed using Dexter theory, as shown in Formula (1) [31]:(1)$${I/x=K[1+\beta (x^{-Q/3})]}^{-1},$$
where K and β are constants, *I* represents the emission intensity at different concentrations, *x* represents the concentration of Tb^3+^, and Q represents the type of interaction. When the values of Q are 6, 8, and 10, the mechanisms for excited state electron transfer in Sm^3+^ correspond to the interactions of electric dipole electric dipole (d-d), electric dipole electric quadrupole (d-q), and electric quadrupole electric quadrupole (q-q), respectively.

Figure 5 illustrates the relationship between the logarithm of fluorescence intensity, log(*I*/*x*), and the logarithm of concentration, log(*x*), for the KYC:0.10Sm^3+^,*x*Tb^3+^ phosphors. As shown in the figure, the slope of the fitted line is −2.4716. The calculated Q value is 7.4148, which is close to 8. The fitting factor *R*^2^ is 0.98763, indicating a good fitting. Therefore, the concentration quenching of the KYC:0.10Sm^3+^,*x*Tb^3+^ system can be attributed to the mechanism of electric dipole-quadrupole interaction.

### 2.4. Energy Level Diagram

Figure 6 shows a schematic diagram of the energy transfer between Tb^3+^ and Sm^3+^. In Tb^3+^ emission, when 374 nm is used as the excitation wavelength, electrons in the ^7^F_6_ ground state of Tb^3+^ are excited to the ^5^G_6_ energy level. Then, some of these excited state electrons will non-radiatively relax to the ^5^D_4_ energy level and eventually return to the ^7^F_6_, ^7^F_5_, ^7^F_4_, and ^7^F_3_ energy levels through radiative transitions. This process results in the emission of photons with wavelengths of 488, 542, 583, and 617 nm [32].

In Sm^3+^ emission, the ground-state electrons in the ^6^H_5/2_ lever are excited to the ^4^L_17/2_ excited state when exposed to 374 nm light. From this excited state, emission occurs at 609, 567, and 558 nm wavelengths, corresponding to the energy transition ^4^G_7/2_ → ^6^H_11/2_, ^4^F_3/2_ → ^6^H_7/2_, and ^4^G_5/2_ → ^6^H_5/2_ [33]. This indicates that electrons in the ^4^L_17/2_ excited state can transition to the ^4^G_7/2_, ^4^F_3/2_, and ^4^G_5/2_ energy levels, emitting photons as they return to the lower energy lever. It is important to note that, in the ^4^G_7/2_ and ^4^F_3/2_ energy levels, only the two transitions (^4^G_7/2_ → ^6^H_11/2_ and ^4^F_3/2_ → ^6^H_7/2_) are capable of converting excited state electrons into photons. As for the ^4^G_5/2_ energy level, when the excited state electrons return to the lower energy levels of ^6^H_5/2_, ^6^H_7/2_, ^6^H_9/2_, and ^6^H_11/2_, they can emit four types of visible light at wavelengths of 558, 596, 642, and 700 nm [15,30]. When the concentration of Sm^3+^ ions is excessively high, cross-relaxation 1, 2, 3, and 4 (CR1, CR2, CR3, CR4) will occur among the Sm^3+^ ions, resulting in concentration quenching [31,34].

In addition, there is an energy transfer between Tb^3+^ and Sm^3+^ ions. When the sample of KYC:0.10Sm^3+^,0.20Tb^3+^ was excited by 488 nm (as shown in Appendix A), several emission peaks can be observed for Sm^3+^ at 609, 567, and 558 nm. This indicates that excited photons at the Tb^3+ 5^D_4_ energy level can be effectively transferred to the ^4^G_7/2_ energy level of Sm^3+^ ions for the emission of Sm^3+^. Another potential pathway for energy transfer is the resonance energy transfer between the ^5^G_6_ level of Tb^3+^ and the ^4^L_17/2_ level of Sm^3+^. The energy levels of ^5^G_6_ (Tb^3+^) and ^4^L_17/2_ (Sm^3+^) are 26,425 cm^−1^ [35] and 26,749 cm^−1^ [10], respectively, with an energy difference of 324 cm^−1^. Excited electrons can transfer energy between similar energy levels through resonance [36]. Therefore, in the KYC:Sm^3+^,Tb^3+^ system, Tb^3+^ can significantly enhance the emission efficiency of Sm^3+^. Additionally, due to the sensitivity of phonon vibrations to temperature, Tb^3+^ and Sm^3+^ co-doped KYC systems can be utilized for temperature sensing.

### 2.5. Temperature Sensing

The temperature-dependent emission spectra of KYC:0.10Sm^3+^,0.20Tb^3+^ phosphors within the temperature range of 300 to 500 K are presented in Figure 7. As the temperature increases, the emission intensity of both Tb^3+^ and Sm^3+^ in the fluorescent powder gradually decreases under 374 nm ultraviolet excitation. The radiative transition from ^5^D_4_ to ^7^F_5_ in Tb^3+^ at 542 nm exhibits the highest intensity but shows the most significant relative decrease. This suggests that the increase in temperature enhances phonon vibrations, thereby promoting the redistribution of excited state electrons. The intrinsic emissions of Sm^3+^ appear at 596 nm and 567 nm, and their emission intensities also demonstrate a thermal quenching effect as temperature rises. Due to the impact of temperature on emission intensity, the temperature-sensing properties of Sm^3+^-Tb^3+^ doped KYC phosphors have been investigated using Fluorescence intensity ratio (FIR) technology.

The relationship between FIR and the temperature of KYC phosphors co-doped with Sm^3+^ and Tb^3+^ can be expressed by Equation (2) [17]:(2)$$\mathrm{FIR}=Ae^{-\frac{∆E}{kT}}+C,$$where A and C are constants, ∆E is the energy gap, and k is the Boltzmann constant.

The absolute sensitivity and relative sensitivity of the sensor can be described using Formulas (3) and (4), respectively.(3)$$S_{A}=\left|\frac{\mathrm{dFIR}}{\mathrm{dT}}\right|=\left|A\frac{∆E}{K_{B}T^{2}}\exp\left(-\frac{∆E}{K_{B}T}\right)\right|,$$(4)$$S_{R}=\left|\frac{1}{\mathrm{FIR}}\frac{\mathrm{dFIR}}{\mathrm{dT}}\right|\times 100\%,$$

Figure 8 shows the thermal sensing characteristic curves of KYC:0.10Sm^3+^ and KYC:0.10Sm^3+^,0.20Tb^3+^. In the experiment, the emission intensities of the ^4^F_3/2_ → ^6^H_7/2_ (567 nm) transition of Sm^3+^, the ^4^G_5/2_ → ^6^H_7/2_ (596 nm) transition of Sm^3+^, and the ^5^D_4_ → ^7^F_5_ (542 nm) transition of Tb^3+^ were labeled and calculated. Among them, ^4^F_3/2_ and ^4^G_5/2_ in Sm^3+^ ions are thermally coupled energy levels [13], while ^4^F_3/2_ in Sm^3+^ ions and ^5^D_4_ in Tb^3+^ ions are non-thermally coupled energy levels. In Figure 8a, the emission intensity ratio (I_567_/I_596_) of the ^4^F_3/2_ and ^4^G_5/2_ thermal coupling energy levels in KYC:0.10Sm^3+^ varies with temperature. The FIR values at various temperatures are fitted using Formula (2). When Sm^3+^ ions are emitted alone, the relationship between FIR and temperature is given by the equation: FIR = 1.37 exp(−1333.96/T) + 0.36. The Adj R-square value is 0.99325, which is close to 1.

Figure 8b presents the absolute and relative sensitivity calculated using Formulas (3) and (4). Both absolute sensitivity and relative sensitivity of KYC:0.10Sm^3+^ increase with rising temperature. The maximum absolute sensitivity observed within the temperature range of 298 K to 498 K is 0.00051 K^−1^ at 498 K, and the maximum relative sensitivity reaches 0.11% at the same temperature. This absolute sensitivity of KYC doped with Sm^3+^ alone is comparable to the maximum absolute sensitivity of 0.00045 K^−1^ (at 750 K) in GdVO4:Sm^3+^ [13]. However, it is lower than that of Sm^3+^-doped oxyfluorotellurite glasses (0.0031 K^−1^) at 700 K [15]. By comparing the temperature values corresponding to the maximum sensitivity, it can be concluded that the KYC:Sm^3+^ system is better suited for measuring relatively low temperatures.

Figure 8c illustrates the thermal sensing characteristics of Tb^3+^ enhanced Sm^3+^ emission. The fitting still utilizes the emission intensity ratio of the ^4^F_3/2_ to ^6^H_7/2_ transition (567 nm) and the ^4^G_5/2_ to ^6^H_7/2_ transition (596 nm) of Sm^3+^. The graph depicts that the variation in the ΔFIR emitted by Sm^3+^ in KYC:0.10Sm^3+^,0.20Tb^3+^ is 0.191, which is greater than the 0.076 observed in KYC:0.10Sm^3+^. The relationship between the FIR and temperature can be expressed with the following equation: FIR = 2.29 exp(−1106.24/T) + 0.16. The Adj R-square is 0.99939. Figure 8d shows the maximum absolute sensitivity of 0.0011 K^−1^ at 492 K, along with a relative sensitivity of 0.34% K^−1^ at 348 K for KYC:0.10Sm^3+^,0.20Tb^3+^. Compared to KYC doped solely with Sm^3+^, the sensitivity values in KYC co-doped with Tb^3+^ and Sm^3+^ are higher, particularly at lower temperatures. This enhancement in sensitivity can be attributed to the increase in temperature, which intensifies phonon vibrations and promotes energy transfer between Tb^3+^ and Sm^3+^.

Figure 8e shows the temperature-dependent relationship of the FIR calculated from the dual emission centers of Tb^3+^ and Sm^3+^ in KYC:0.10Sm^3+^,0.20Tb^3+^. Among them, FIR refers to the emission intensity ratio of the Sm^3+^ transition from ^4^F_3/2_ to ^6^H_7/2_ (567 nm) to the Tb^3+^ transition from ^5^D_4_ to ^7^F_5_ (542 nm). The formula derived for FIR within the temperature range of 298 K to 498 K is as follows: FIR = 2.15 exp(−1146.74/T) + 0.10. Additionally, in Figure 8f, the maximum relative and absolute sensitivity are observed at 0.00099 K^−1^ (498 K) and 0.42% K^−1^ (340 K), respectively. The sensitivity value is greatly enhanced in fluorescent powder with dual emission centers.

Figure 9 shows that the value of the FIR decreases with increasing temperature in the KYC:0.10Sm^3+^,0.20Tb^3+^ sample. The FIR is calculated using the ratio of the emission intensity of Tb^3+^ at 542 nm to that of Sm^3+^ at 567 nm. As shown in Figure 9a, the relationship between FIR and temperature can be well fitted by the following equation: FIR = 89.49 exp(−28.61/T) − 91.62. The Adj. R-square for FIR data is close to 1, indicating a strong fit and suggesting that this model accurately represents the relationship between FIR and temperature. In Figure 9b, the absolute and relative sensitivity are the highest at −0.31 K^−1^ (298 K) and −0.46%K^−1^ (298 K), respectively. Comparing this with Figure 8, the maximum sensitivity value can be achieved by choosing the optimal data processing method, even for the same material. This indicates that the data processing technique is crucial for fluorescence temperature sensing. Furthermore, Stefanska et al. developed a novel method that employs the opposing thermal reactions of rare earth ions in Ground State Absorption (GSA) and Excited State Absorption (ESA) to enhance the application of Tb^3+^ ions in high-sensitivity thermometry [37]. Therefore, exploring various data processing methods is beneficial for improving the potential applications of fluorescent powders in temperature sensing.

Table 2 summarizes the sensitivity values obtained from the Sm^3+^ emission in various host materials. A comparison of the absolute sensitivity values indicates that the maximum sensitivities observed in the GdVO_4_ [13], YNbO_4_ [12], and oxyfluorotellurite glasses [15] hosts are comparable to those found in the current KYC:Sm^3+^. However, the temperature at which the maximum absolute sensitivity occurs in this KYC host has decreased by 200 K compared to the conventional host. This decrease indicates that the current system is more suitable for temperature measurements in relatively low-temperature environments. When Tb^3+^ ions are introduced into the KYC:0.10Sm^3+^, the maximum absolute sensitivity values for KYC:0.10Sm^3+^,0.20Tb^3+^ increase two times to 0.0011 K^−1^. The maximum relative sensitivity rises to 0.34% K^−1^, with the corresponding temperature measurement point dropping to 348 K. By using the emission intensities at 567 nm (Sm^3+^) and 542 nm (Tb^3+^) as the FIR, the maximum relative sensitivity has increased to 0.42% K^−1^. When we reverse the ratio of these two emission intensities with each other, the FIR is set to the ratio of 542 nm and 567 nm. The absolute values of absolute sensitivity and relative sensitivity reach 0.031 K^−1^ and 0.46% K^−1^, respectively. Therefore, the KYC:0.10Sm^3+^,0.20Tb^3+^ is more appropriate for temperature measurement at room temperature in the higher sensitivity values.

Figure 10 presents the thermal cycling schematic for the KYC:0.10Sm^3+^,0.20Tb^3+^ fluorescent powder. The graph demonstrates that the fluorescence intensity ratio of I_542_/I_567_ remains relatively stable over five consecutive heating and cooling cycles. The tests were performed at three elevated temperatures to evaluate the thermal stability of the sample: 375 K, 425 K, and 475 K. At each of these temperatures, the ratio could be fitted with a straight line, indicating strong thermal stability. As a result, when the fluorescent powder is operated at a lower temperature, its reliability and potential for application are significantly enhanced.

## 3. Materials and Methods

KY_0.9-*x*_(CO_3_)_2_:0.10Sm^3+^,*x*Tb^3+^(labeled as KYC:Sm^3+^,*x*Tb^3+^) samples were prepared via hydrothermal method. Firstly, 1.9827g of K_2_CO_3_·1.5H_2_O was weighed and placed into a beaker. Then, 20 mL of deionized water was added, and the mixture was thoroughly stirred using a magnetic stirrer. To prepare the target sample of KYC:Sm^3+^,Tb^3+^ (1 mmol), Y(NO_3_)_3_∙6H_2_O (99.99%), Sm(NO_3_)_3_∙6H_2_O (99.99%), and Tb(NO_3_)_3_∙6H_2_O (99.99%) were weighed and dissolved in 3 mL of deionized water. The mixed solution was then added dropwise to the K_2_CO_3_ solution, and the initial pH value of the solution was 11.4. After stirring for 30 min, the pH was adjusted to 9.5 with nitric acid. After thorough mixing, the reaction solution was transferred into the reaction kettle and heated at 200 °C for 8 h. After cooling, the sample was washed three times with ethanol and deionized water and then dried at 80 °C for 1 h. A new fluorescent powder KYC:Sm^3+^,Tb^3+^ was obtained. All the KY_1-*x*_(CO_3_)_2_:*x*Sm^3+^ (KYC:*x*Sm^3+^) samples were prepared using the same method.

The crystal structures of KYC:Sm^3+^ and KYC:Sm^3+^,Tb^3+^ were analyzed using an X-ray diffractometer with a scanning range of 10–80° (PANalytic, Almelo, The Netherlands). The morphologies and energy dispersive spectrum of KYC:Sm^3+^,Tb^3+^ were imaged via cold field emission scanning electron microscopy (Regulus 8220, Hitachi High-Tech Co., Tokyo, Japan). Spectral characteristics were analyzed using an FLS920 fluorescence spectrometer with a xenon lamp as the light source (Edinburgh Instruments, Livingston, UK). The scanning range of the emission spectrum and excitation spectrum is 200–800 nm, and the slit width of the excitation grating and emission grating are 1.5 nm and 0.2 nm, respectively. Temperature-dependent emissions were recorded using an Optical Multichannel Analyzer (SP2556, Princeton Instruments, Trenton, NJ, USA) with a 374 nm laser excitation source. The temperature varied from room temperature to 500 K and was controlled by a temperature controller (LC-DB-AB, LICHEN, Shanghai, China). Cell parameters were fitted by Rietveld refinement using FullProf software (5.10).

## 4. Conclusions

Tb^3+^-doped KYC:Sm^3+^ was synthesized via the hydrothermal method. The thermal quenching phenomenon of Sm^3+^ is used for thermometry. In the KYC host, Sm^3+^ achieves maximum sensitivity at relatively low temperatures. The optimum doping concentration of Sm^3+^ in KYC:*x*Sm^3+^ phosphor was 0.1. A pair of thermal coupling energy levels, ^4^F_3/2_ and ^4^G_5/2_, in Sm^3+^ have been adopted, and the maximum absolute sensitivity has been calculated to be 0.00051 K^−1^ at 498 K. Notably, the temperature at which the maximum sensitivity occurs is approximately 200 K lower than that of other samarium-doped phosphors. Moreover, the energy transfer between Tb^3+^ and Sm^3+^ can further enhance the sensitivity of Sm^3+^ to temperature changes. Additionally, different data processing methods lead to varying values of thermal sensitivities. The maximum absolute and relative sensitivities of KYC:0.10Sm^3+^,0.20Tb^3+^, obtained using conventional FIR techniques, are 0.00099 K^−1^ and 0.42% K^−1^. However, when using the specific FIR processing method that decreases with increasing temperature, the maximum absolute and relative sensitivities of KYC:0.10Sm^3+^,0.20Tb^3+^ are −0.031 K^−1^ and −0.46% K^−1^. The temperature point for maximum sensitivity is significantly lower than that of conventional Sm^3+^ thermosensitive materials, making KYC:Sm^3+^,Tb^3+^ suitable for temperature measurements in environments from room temperature up to 200 °C.

## Figures and Tables

**Figure 1 molecules-30-00767-f001:**
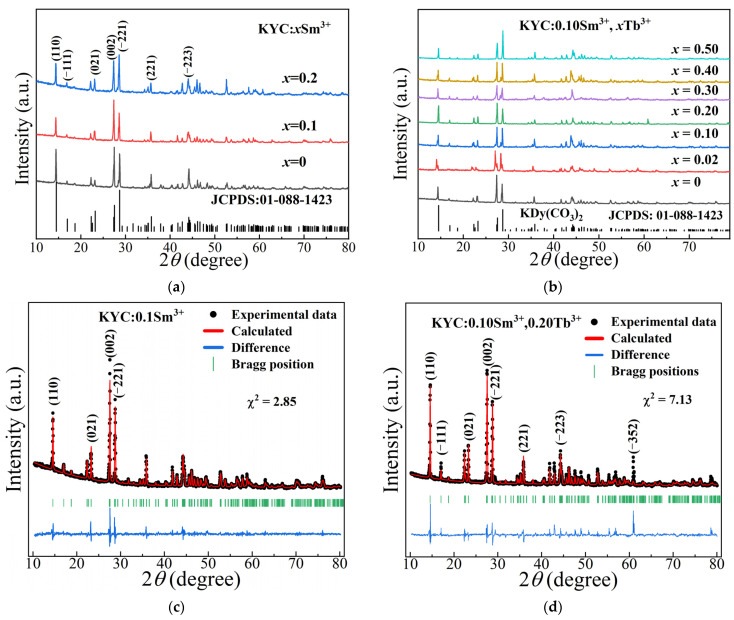
XRD patterns of (**a**) KYC:*x*Sm^3+^; (**b**) KYC:0.10Sm^3+^,*x*Tb^3+^; Rietveld refinements of (**c**) KYC:0.1Sm^3+^; (**d**) KYC:0.10Sm^3+^,0.20Tb^3+^.

**Figure 2 molecules-30-00767-f002:**
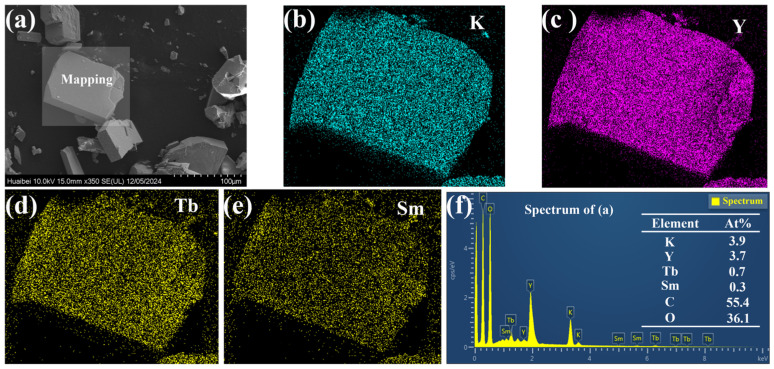
(**a**) Morphologies of KYC:0.10Sm^3+^,0.20Tb^3+^; (**b**–**e**) elements distribution; (**f**) energy dispersive spectrum of crystal.

**Figure 3 molecules-30-00767-f003:**
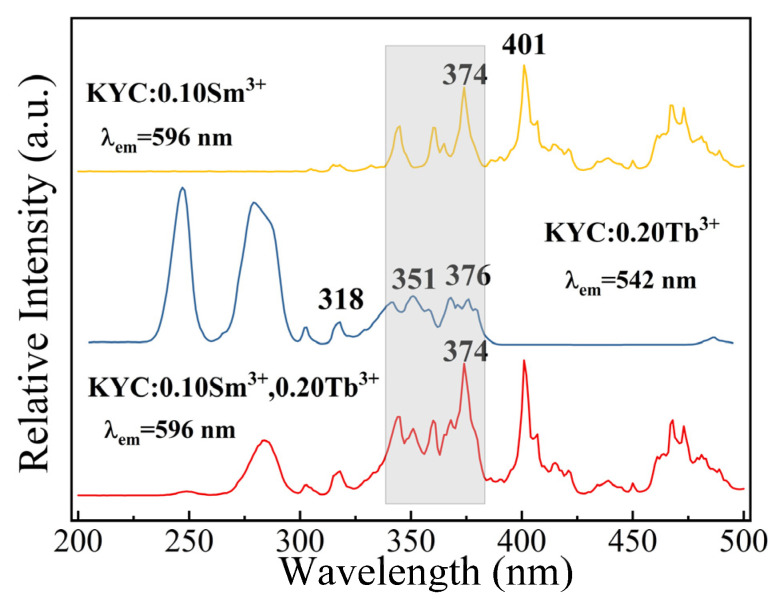
Excitation spectra of KYC:0.10Sm^3+^, KYC:0.20Tb^3+^, and KYC:0.10Sm^3+^,0.20Tb^3+^.

**Figure 4 molecules-30-00767-f004:**
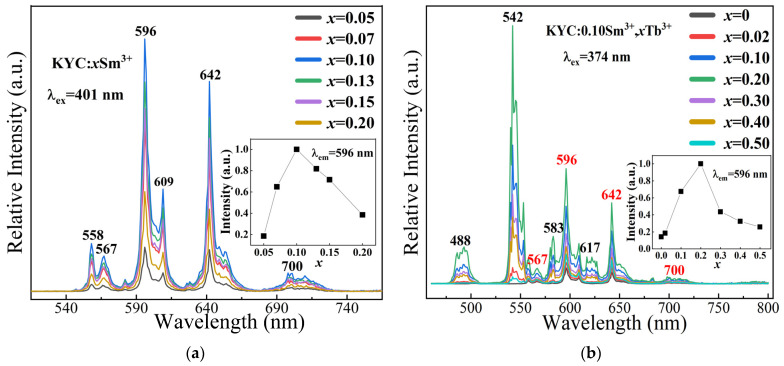
Emission spectra of (**a**) KYC:*x*Sm^3+^; (**b**) KYC:0.10Sm^3+^,*x*Tb^3+^.

**Figure 5 molecules-30-00767-f005:**
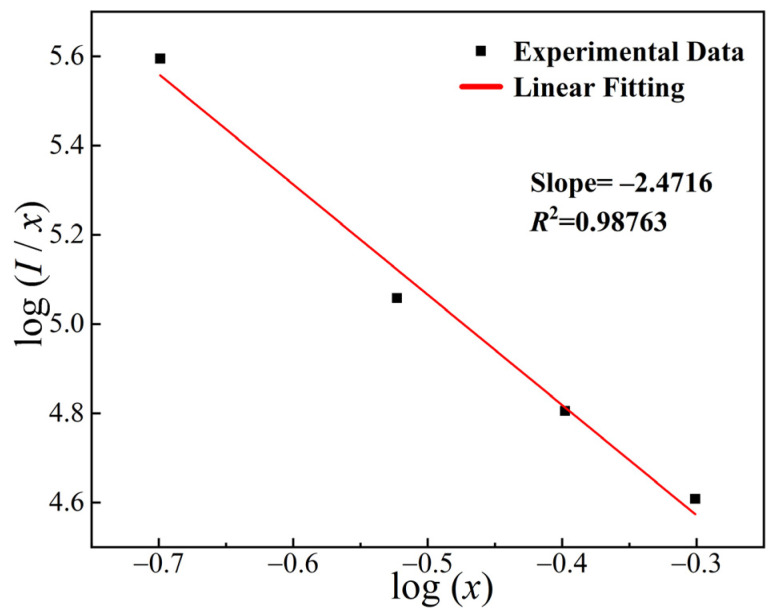
The relationship between log(*I*/*x*) and log(*x*) of KYC:0.10Sm^3+^,*x*Tb^3+^ (*x* = 0.20, 0.30, 0.40, 0.50).

**Figure 6 molecules-30-00767-f006:**
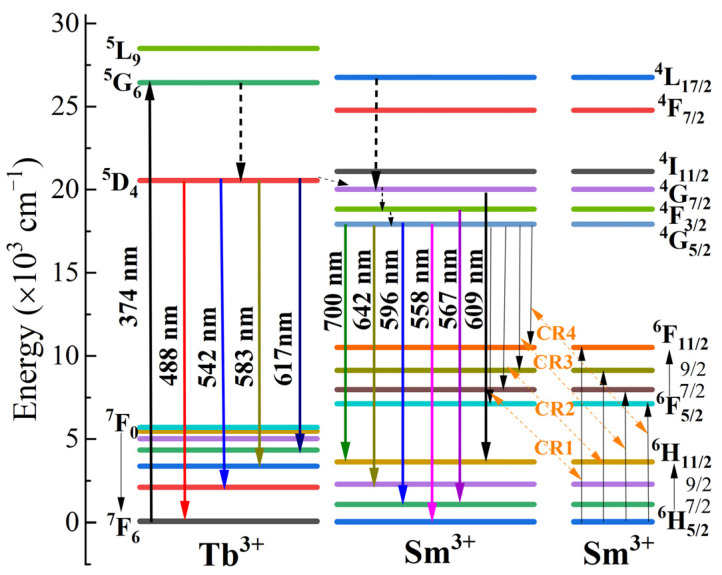
Energy level diagrams of KYC:Sm^3+^,Tb^3+^ (Dash line: electron transfer process; Solid line: Excitation or emission process; Arrow: transfer direction).

**Figure 7 molecules-30-00767-f007:**
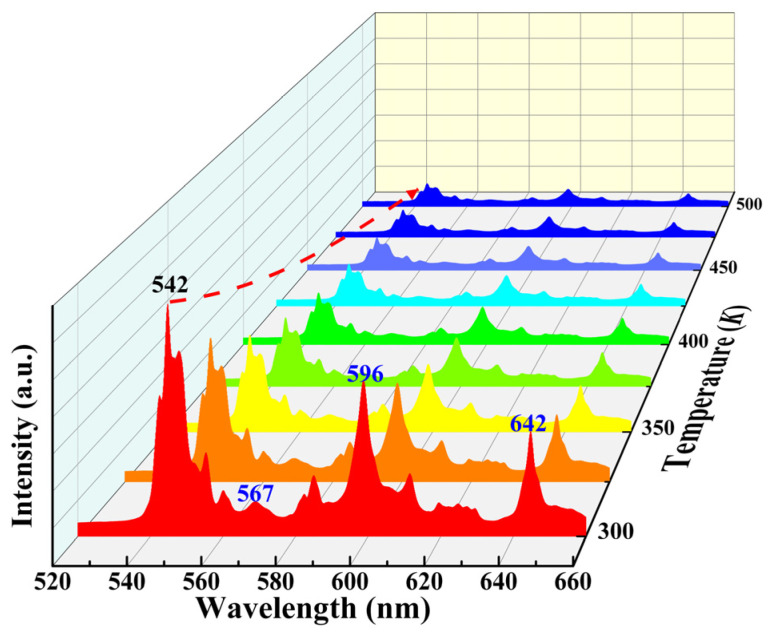
Temperature-dependent emission spectra of KYC:0.10Sm^3+^,0.20Tb^3+^.

**Figure 8 molecules-30-00767-f008:**
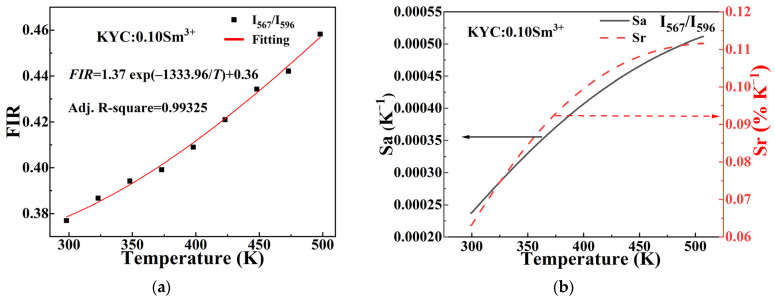
Temperature-dependent curves of FIR, absolute sensitivity (Sa) and relative sensitivity (Sr): (**a**) FIR (I_567_/I_596_) and (**b**) sensitivity of KYC:0.10Sm^3+^; (**c**) FIR (I_567_/I_596_) and (**d**) sensitivity of KYC:0.10Sm^3+^,0.20Tb^3+^; (**e**) FIR (I_567_/I_542_) and (**f**) sensitivity of KYC:0.10Sm^3+^,0.20Tb^3+^.

**Figure 9 molecules-30-00767-f009:**
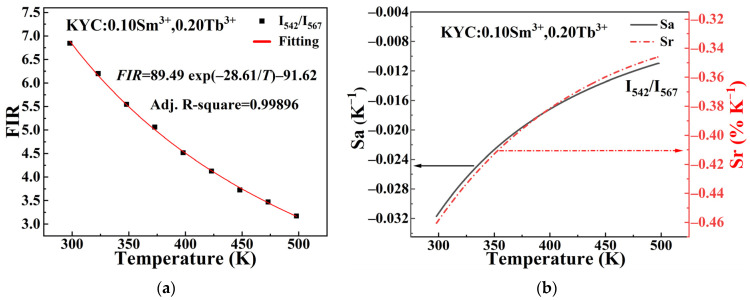
Temperature-dependent curves of KYC:0.10Sm^3+^,0.20Tb^3+^ (**a**) FIR (I_542/_I_567_); (**b**) absolute sensitivity (S_a_) and relative sensitivity (S_r_).

**Figure 10 molecules-30-00767-f010:**
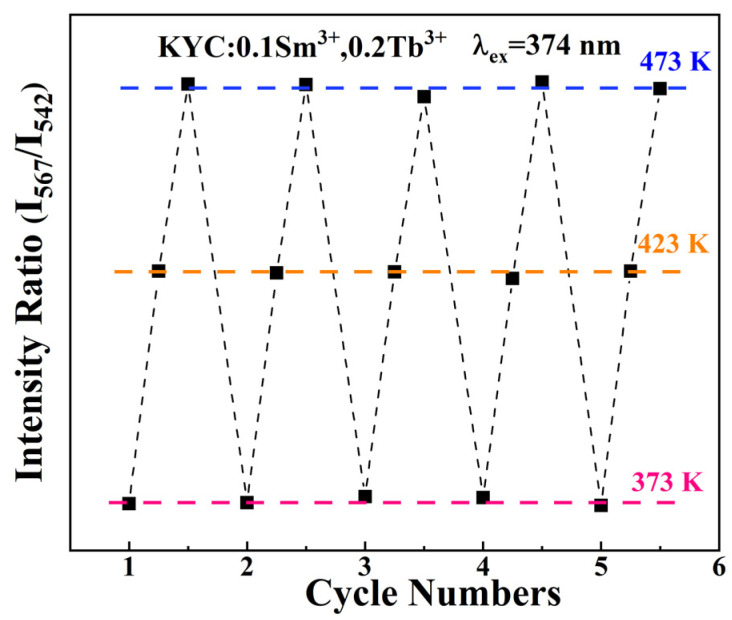
Thermal cycles of KYC:0.10Sm^3+^,0.20Tb^3+^.

**Table 1 molecules-30-00767-t001:** Refined parameters of typical materials.

Parameters	KY(CO_3_)_2_ [21]	KYC:0.1Sm^3+^	KYC:0.10Sm^3+^,0.20Tb^3+^
Crystal System	Monoclinic	Monoclinic	Monoclinic
Space group	C2/c	C2/c	C2/c
a (Å)	8.488	8.516	8.510
b (Å)	9.442	9.464	9.462
c (Å)	6.913	6.934	6.932
α (°)	90.00	90.00	90.00
β (°)	110.96	111.03	111.02
γ (°)	90.00	90.00	90.00
Cell Volume (Å^3^)	517.373	521.625	521.032

**Table 2 molecules-30-00767-t002:** Maximum Sensitivity Values of Different Fluorescent Materials.

Materials	FIR (nm/nm)	S_a-max_ (K^−1^)	S_r-max_ (% K^−1^)	References
GdVO_4_:Sm^3+^	533 nm/564 nm	0.00045@750 K	-	[13]
YNbO_4_:Sm^3+^	531 nm/566 nm	0.0007@700 K	0.43@500 K	[12]
Glasses:Sm^3+^	563 nm/599 nm	0.0031@700 K	-	[15]
KYC:0.10Sm^3+^	567 nm/596 nm	0.00051@498 K	0.11@498 K	This work
KYC:0.10Sm^3+^,0.20Tb^3+^	567 nm/596 nm	0.0011@492 K	0.34@348 K	This work
KYC:0.10Sm^3+^,0.20Tb^3+^	567 nm/542 nm	0.00099@498 K	0.42@340 K	This work
KYC:0.10Sm^3+^,0.20Tb^3+^	542 nm/567 nm	−0.031@298 K	−0.46@298 K	This work

## Data Availability

Data is contained within the article (and Appendix A).

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
