# Peer review of "Sensitivity Enhancement of Thermometry in Tb3+-Doped KY(CO3)2:Sm3+ by Energy Transfer"

_molecules, 2025, doi:10.3390/molecules30040767_

Round 1
Reviewer 1 Report
Comments and Suggestions for Authors
Dear Authors,
I have reviewed the manuscript you submitted, which investigates the properties of KY(CO3)2 doped with samarium and co-doped with both terbium and samarium. The manuscript includes results on the structure, microstructure, luminescence, and temperature-dependent emission spectra. Despite your evident efforts to provide detailed information, there are areas that still need improvement. Below are my recommendations and comments:
1. In Section 3, "Materials and Methods," you mention that the pH of the solution was adjusted using nitric acid; however, the initial pH value is not provided. Please include this information and clarify why pH correction was necessary.
2. The descriptions of the samples do not indicate the units for the dopant additions, Sm and Tb, in KYC:xSm3+ (x=0, 0.1, 0.2) and 77 KYC:0.10Sm3+, xTb3+ (x=0, 0.01, 0.10, 0.20, 0.30, 0.40, 0.50). Please specify whether these are in mol% or wt%.
3. In Section 2.1, please clarify the statement on lines 85-86 regarding the use of Sm3+ as a doping ion. You mention that its concentration in KYC can reach 20% (see Figure 1a). Additionally, you state that KYC doped with Sm3+ and Tb3+ can maintain lattice stability even with Tb3+ concentrations as high as 50% (see Figure 1b). Please elaborate on this point.
4. Please include information about the parameters of the crystal lattice as well as the positions of the main peaks in the X-ray diffraction patterns in Section 2.1.
5. Regarding your comment: "Figures 1a and 1b show that the main diffraction peaks for KYC:Sm3+ and KYC:Sm3+,Tb3+ are located at similar positions, with only minor differences in peak intensity." It is clear that not all peak positions match exactly with the provided JCPDS map. However, samples KYC:Sm3+ (0.2) and KYC:Sm3+,Tb3+ (0.5) demonstrate improved crystallinity compared to other analyzed samples. The presence of several broad and diffuse peaks in the X-ray diffraction patterns indicates less-than-ideal crystallinity, which likely depends on the roles of the alloying components—namely, whether they substitute yttrium ions or occupy intermediate positions in the crystal lattice, as well as their concentration.
6. In Section 3, please provide details about the FLS920 spectrometer, specifically whether it is equipped with sample temperature control. If it is not, please describe how fluorescence measurements were performed at different temperatures.
Best regards,
Comments on the Quality of English LanguageDear Authors,
The quality of the English language can be improved in the present manuscript. Please check it for grammatical and meaning errors.
Author Response
Comments 1: In Section 3, "Materials and Methods," you mention that the pH of the solution was adjusted using nitric acid; however, the initial pH value is not provided. Please include this information and clarify why pH correction was necessary.
Response 1: Thanks for your kind reminder. We have added corresponding descriptions in the article.
"The mixed solution was then added dropwise to the K2CO3 solution, and the initial pH value of the solution was 11.4. "
Additionally, we have conducted detailed research on KYC doped with Tb3+. Our findings indicate that the optimal pH range for preparing KYC:Tb3+ samples is between 8.5 and 10.5, as detailed in our patent: CN202011189803.8. If the pH of the reaction solution is below 8.5 or above 10.5, impure phases could form in the final products. Thus, selecting a pH of 9.5 allows us to obtain KYC:Sm3+,Tb3+ with good crystallinity and a pure phase.
Comments 2: The descriptions of the samples do not indicate the units for the dopant additions, Sm and Tb, in KYC:xSm3+ (x=0, 0.1, 0.2) and 77 KYC:0.10Sm3+, xTb3+ (x=0, 0.01, 0.10, 0.20, 0.30, 0.40, 0.50). Please specify whether these are in mol% or wt%.
Response 2: Thank you for your reminder.
The terms "KYC:xSm3+" and "KYC:0.10Sm3+,xTb3+" were unclear and could easily be misunderstood. Therefore, we have revised them to "KY1-x(CO3)2:xSm3+" and "KY0.9-x(CO3)2:0.10Sm3+,xTb3+". This new formulation is clearer in meaning. For instance, "KYC:0.1Sm3+" indicates that the concentration of sm3+ in the KYC accounts for 10 mol%. The revised text is as follows:
"The X-ray diffraction (XRD) patterns of KY1-x(CO3)2:xSm3+ (x=0, 0.1, 0.2) and KY0.9-x(CO3)2:0.10Sm3+,xTb3+ (x=0, 0.02, 0.10, 0.20, 0.30, 0.40, 0.50) are illustrated in Figure 1. Figures 1a and 1b show that the main diffraction peaks for KYC:Sm3+ and KYC:Sm3+,Tb3+ are located at similar positions, with only minor differences in peak intensity. The positions of diffraction peaks in the fluorescent powder align with the standard card (JCPDS: 01-088-1423), and no additional diffraction peaks were detected. The reason is that Tb3+ (1.040 Å), Sm3+ (1.079 Å), and Y3+ (1.019 Å) are all trivalent ions with comparable ionic radii [25]. Tb3+ and Sm3+ ions can easily substitute for Y3+ ions in KYC to form the monoclinic phase KYC:Sm3+ and KYC:Sm3+,Tb3+. When Sm3+ is used alone as a doping ion, its concentration in KYC:0.20Sm3+ can reach 20 mol% (Figure 1a). When Tb3+ ions are co-doped in KYC:0.10Sm3+, the concentration of Tb3+ can reach 50 mol%. KYC:0.10Sm3+,0.50Tb3+ can still maintain its original monoclinic structure (Figure 1b)."
Comments 3: In Section 2.1, please clarify the statement on lines 85-86 regarding the use of Sm3+ as a doping ion. You mention that its concentration in KYC can reach 20% (see Figure 1a). Additionally, you state that KYC doped with Sm3+ and Tb3+ can maintain lattice stability even with Tb3+ concentrations as high as 50% (see Figure 1b). Please elaborate on this point.
Response 3: Thank you for your constructive suggestions. We have modified the corresponding description in the text.
"When Sm3+ is used alone as a doping ion, its concentration in KYC:0.20Sm3+ can reach 20 mol% (Figure 1a). When Tb3+ ions are co-doped in KYC:0.10Sm3+, the concentration of Tb3+ can reach 50 mol%. KYC:0.10Sm3+,0.50Tb3+ can still maintain its original monoclinic structure (Figure 1b)."
Comments 4: Please include information about the parameters of the crystal lattice as well as the positions of the main peaks in the X-ray diffraction patterns in Section 2.1.
Response 4: Thank you for your constructive suggestions. We have added the Rietveld refinement of XRD data and discussed it accordingly.
"Figure 1c and 1d illustrate the Rietveld refinement results for KYC:0.1Sm3+ and KYC:0.10Sm3+,0.20Tb3+ analyzed using FullProf software. The figures show that most diffraction peaks are well-fitted, although there are a few exceptions. In Figure 1c, the (002) and (–221) crystal planes show discrepancies, while in Figure 1d, the (–111) and (–352) crystal planes are less accurately fitted. Furthermore, the fitting quality for the diffraction peaks in Figure 1c is superior to that in Figure 1d. The corresponding fitting parameters are summarized in Table 1. The fitted cell volume reveals that KYC:0.1Sm3+ has a greater value than KYC:0.10Sm3+,0.20Tb3+. This difference is primarily due to the fact that the radius of Sm3+ is larger than that of Tb3+, which is, in turn, larger than that of Y³⁺."
(Figures 1c and 1d have been inserted into the article)
Figure 1. XRD patterns of (a) KYC:xSm3+; (b)KYC:0.10Sm3+,xTb3+; Rietveld refinements of (c) KYC:0.1Sm3+; (d) KYC:0.10Sm3+,0.20Tb3+
Table 1. Refined parameters of typical materials
Parameters |
KY(CO3)2 [21] |
KYC:0.1Sm3+ |
KYC:0.10Sm3+,0.20Tb3+ |
Crystal System |
Monoclinic |
Monoclinic |
Monoclinic |
Space group |
C2/c |
C2/c |
C2/c |
a (Å) |
8.488 |
8.516 |
8.510 |
b (Å) |
9.442 |
9.464 |
9.462 |
c (Å) |
6.913 |
6.934 |
6.932 |
a (°) |
90.00 |
90.00 |
90.00 |
b (°) g (°) Cell Volume (Å3) |
110.96 |
111.03 |
111.02 |
90.00 |
90.00 |
90.00 |
|
517.373 |
521.625 |
521.032 |
Comments 5: Regarding your comment: "Figures 1a and 1b show that the main diffraction peaks for KYC:Sm3+ and KYC:Sm3+,Tb3+ are located at similar positions, with only minor differences in peak intensity." It is clear that not all peak positions match exactly with the provided JCPDS map. However, samples KYC:Sm3+ (0.2) and KYC:Sm3+,Tb3+ (0.5) demonstrate improved crystallinity compared to other analyzed samples. The presence of several broad and diffuse peaks in the X-ray diffraction patterns indicates less-than-ideal crystallinity, which likely depends on the roles of the alloying components—namely, whether they substitute yttrium ions or occupy intermediate positions in the crystal lattice, as well as their concentration.
Response 5: Thank you for your explanation and guidance. As you mentioned, the change in peak shape is influenced by various factors. The alteration in the shape and intensity of the diffraction peaks is partially attributed to the presence of large grains and doping effects. SEM images revealed that the grains are large and prone to fracturing. Additionally, XRD data fitting indicated that the substitution doping of terbium (Tb) and samarium (Sm) altered the size of the cell volume.
Comments 6: In Section 3, please provide details about the FLS920 spectrometer, specifically whether it is equipped with sample temperature control. If it is not, please describe how fluorescence measurements were performed at different temperatures.
Response 6: Thank you for pointing this out. We added the description of the temperature control instrument.
"Temperature-dependent emissions were recorded using an Optical Multichannel Analyzer (SP2556, Princeton Instruments, Trenton, US) with a 374 nm laser excitation source. The temperature was varied from room temperature to 500 K and controlled by a Temperature Controller (LC-DB-AB, LICHEN, Shanghai, China). Cell parameters were fitted by Rietveld refinement using FullProf software (5.10)."
Reviewer 2 Report
Comments and Suggestions for Authors
Before publishing the manuscript, it is necessary to make the following corrections.
1. In the introduction, it is important to mention the application of the Tb3+ doped 2 KYC:Sm3+ materials to highlight the importance of the research.
2. Explain the novelty of the research with what has already been reported, for example, in the following article.
https://pubs.acs.org/doi/10.1021/acs.jpcc.0c11631
3. In the experimental section, why did the variation of Tb3+ go from x=0.01 to x =0.50? How were these values chosen?
4. Expand the conclusions based on the results.
Author Response
Comments 1: In the introduction, it is important to mention the application of the Tb3+ doped KYC:Sm3+ materials to highlight the importance of the research.
Response 1: Thank you for your constructive suggestions. We added some discussion about the advantages of KYC.
"Tb3+ doped KYC has demonstrated a fluorescence quantum efficiency of up to 177% [22]. The efficient luminescence of Tb3+ ions is expected to enhance the fluorescence intensity ratio of dual ion emission to obtain larger temperature sensitivity values. Additionally, Tb3+ and Sm3+ ions can facilitate effective energy transfer due to their similar energy levers [23, 24]. A smaller difference in energy levels results in a greater influence of phonons on energy transfer at the same temperature. The interaction between Sm3+ and Tb3+ in KYC could significantly improve the temperature sensitivity of sm3+, especially at room temperature. Therefore, this work incorporates Tb3+ and Sm3+ ions into the KYC system to explore the spectral characteristics, energy transfer, thermal quenching mechanisms, and temperature sensing properties of KYC:Sm3+,Tb3+."
Comments 2: Explain the novelty of the research with what has already been reported, for example, in the following article. https://pubs.acs.org/doi/10.1021/acs.jpcc.0c11631
Response 2: Thanks for your constructive suggestions. Some novel methods are essential for enhancing the use of phosphors in thermometry applications.
"In Figure 9b, the absolute and relative sensitivity are the highest at –0.31 K⁻1 (298 K) and –0.46%K⁻1 (298 K), respectively. Comparing this with Figure 8, the maximum sensitivity value can be achieved by choosing the optimal data processing method, even for the same material. This indicates that the data processing technique is crucial for fluorescence temperature sensing. Furthermore, Stefanska et al. developed a novel method that employs the opposing thermal reactions of rare earth ions in Ground State Absorption (GSA) and Excited State Absorption (ESA) to enhance the application of Tb³⁺ ions in high-sensitivity thermometry [37]. Therefore, exploring various data processing methods is beneficial for improving the potential applications of fluorescent powders in temperature sensing."
[37] Stefanska, J.; Chrunik, M.; Marciniak, L., Sensitivity Enhancement of the Tb3+-Based Single Band Ratiometric Luminescent Thermometry by the Metal-to-Metal Charge Transfer Process. Phys. Chem. C 2021, 125(9), 5226-5232.
Comments 3: In the experimental section, why did the variation of Tb3+ go from x=0.01 to x =0.50? How were these values chosen?
Response 3: We would like to thank the reviewer for highlighting the data inconsistency in the text. In the X-ray Diffraction (XRD) analysis shown in Figure 1, we initially used x=0.01 to represent the diffraction pattern of the sample doped with Tb in order to observe the changes before and after doping. However, during the fluorescence spectrum test, we observed that the intensity enhancement for KYC:0.01Sm3+ was small, making it difficult to distinguish between the samples at x=0 and x=0.01. As a result, we decided to use the sample at x=0.02, which led to inconsistencies in the data.
To resolve this issue, we have decided to replace the XRD diffraction pattern for x=0.01 in the original Figure 1 with the diffraction pattern for the x=0.02 sample. This change will ensure consistency across the data presented in the text.
"The X-ray diffraction (XRD) patterns of KY1-x(CO3)2:xSm3+ (x=0, 0.1, 0.2) and KY0.9-x(CO3)2:0.10Sm3+,xTb3+ (x=0, 0.02, 0.10, 0.20, 0.30, 0.40, 0.50) are illustrated in Figure 1. Figures 1a and 1b show that the main diffraction peaks for KYC:Sm3+ and KYC:Sm3+,Tb3+ are located at similar positions, with only minor differences in peak intensity. "
Figure 1. XRD patterns of (a) KYC:xSm3+; (b)KYC:0.10Sm3+,xTb3+
Comments 4: Expand the conclusions based on the results.
Response 4: Thank you for your constructive suggestions. We added some experimental results to highlight the advantages of KYC host.
"Tb3+-doped KYC:Sm3+ was synthesized via the hydrothermal method. The thermal quenching phenomenon of Sm3+ is used for thermometry. In the KYC host, Sm3+ achieves maximum sensitivity at relatively low temperatures. The optimum doping concentration of Sm3+ in KYC:xSm3+ phosphor was 0.1. A pair of thermal coupling energy levels, 4F3/2 and 4G5/2, in Sm3+ have been adopted, and the maximum absolute sensitivity has been calculated to be 0.00051 K⁻¹ at 498 K. Notably, the temperature at which the maximum sensitivity occurs is approximately 200 K lower than that of other samarium-doped phosphors."